# MAIT cells contribute to protection against lethal influenza infection in vivo

Bonnie van Wilgenburg [1,2], Liyen Loh[1], Zhenjun Chen [1], Troi J. Pediongco[1], Huimeng Wang [1], Mai Shi[1], Zhe Zhao[1], Marios Koutsakos [1], Simone Nüssing[1], Sneha Sant[1], Zhongfang Wang[1], Criselle D'Souza [1], Xiaoxiao Jia[1], Catarina F. Almeida[1,3], Lyudmila Kostenko[1], Sidonia B.G. Eckle [1], Bronwyn S. Meehan[1], Axel Kallies[1], Dale I. Godfrey [1,3], Patrick C. Reading[1], Alexandra J. Corbett [1], James McCluskey [1], Paul Klenerman[2], Katherine Kedzierska[1] & Timothy S.C. Hinks [1,4]

Mucosal associated invariant T (MAIT) cells are evolutionarily-conserved, innate-like lymphocytes which are abundant in human lungs and can contribute to protection against pulmonary bacterial infection. MAIT cells are also activated during human viral infections, yet it remains unknown whether MAIT cells play a significant protective or even detrimental role during viral infections in vivo. Using murine experimental challenge with two strains of influenza A virus, we show that MAIT cells accumulate and are activated early in infection, with upregulation of CD25, CD69 and Granzyme B, peaking at 5 days post-infection. Activation is modulated via cytokines independently of MR1. MAIT cell-deficient MR1$^{-/-}$ mice show enhanced weight loss and mortality to severe (H1N1) influenza. This is ameliorated by prior adoptive transfer of pulmonary MAIT cells in both immunocompetent and immunodeficient RAG2$^{-/-}$γC$^{-/-}$ mice. Thus, MAIT cells contribute to protection during respiratory viral infections, and constitute a potential target for therapeutic manipulation.

[1] Department of Microbiology and Immunology, The Peter Doherty Institute for Infection and Immunity, The University of Melbourne, Parkville, VIC 3000, Australia. [2] Peter Medawar Building for Pathogen Research and Translational Gastroenterology Unit, Nuffield Department of Clinical Medicine, University of Oxford, Oxford OX3 3SY, UK. [3] ARC Centre of Excellence in Advanced Molecular Imaging, The University of Melbourne, Parkville, VIC 3010, Australia. [4] Respiratory Medicine Unit, Nuffield Department of Medicine, Experimental Medicine Division, University of Oxford, Oxford OX3 9DU, UK. These authors contributed equally: Bonnie van Wilgenburg, Liyen Loh. These authors jointly supervised this work: James McCluskey, Paul Klenerman, Katherine Kedzierska, Timothy S. C. Hinks.  Correspondence and requests for materials should be addressed to P.K. (email: paul.klenerman@medawar.ox.ac.uk)

     1

Mucosal associated invariant T (MAIT) cells are an abundant and evolutionarily-conserved[1] class of innate-like lymphocytes[2], which comprise up to 10% of human peripheral blood[3] and respiratory mucosal T cells[4,5]. MAIT cell development and antigen (Ag)-specific activation are restricted by the monomorphic β2-microglobulin-associated molecule MHC related protein-1 (MR1)[6]. MAIT cells express a semi-invariant αβ T cell receptor (TCR)[7] shown to recognise metabolic derivatives of highly-conserved riboflavin biosynthetic pathways[8,9], that are expressed by a wide range of bacteria, mycobacteria and yeasts[10]. These molecules are usually short-lived, but are stabilised by MR1, and include 5-(2-oxopropylideneamino)-6-D-ribitylaminouracil (5-OP-RU), which is a potent MAIT cell ligand[11], and can be loaded onto MR1 to form specific tetramers to track human and murine MAIT cells[9,12]. Consistent with these observations, we and others have shown that MAIT cells contribute to protection in vivo against intracellular bacteria including Francisella tularensis[13], Legionella longbeachae[14] and mycobacteria[15], whilst human in vitro data suggest they may contribute also to protection against a variety of other respiratory bacterial pathogens, including Haemophilus influenzae[5,16] and Streptococcus pneumoniae[17,18].

Although this MR1-TCR dependent activation appears specific for riboflavin-expressing prokaryotic pathogens, there is emerging evidence that MAIT cells have the potential also to be involved in immune responses to viruses. MAIT cells have constitutively high surface expression of the interleukin (IL)-18 receptor and IL-12 receptor[19], and we have shown they can be activated by IL-18 synergistically with IL-12[20] or IL-15 or type I interferons (IFN)[21] to induce expression of IFN-γ and Granzyme B[22]. These responses are MR1 and TCR independent[20,21]. These phenotypically innate features of MAIT cells, in common with invariant natural killer T cells (iNKT), are believed to be driven by expression of the master transcription factor promyelocytic leukaemia zinc finger protein (PLZF)[19,23].

We have previously shown that peripheral MAIT cells are activated in vivo during human infections with influenza A virus (IAV), dengue virus and hepatitis C, whilst in vitro these viruses activate MAIT cells in an IL-18 dependent manner[21]. Moreover, we showed that activation correlated with disease activity in IAV infection[21] and reduced peripheral blood MAIT cell frequencies were associated with death in severe avian H7N9 IAV disease[22]. Activation has also been observed by others in hepatitis B[24], hepatitis C[25] and Zika[26] infections, whilst reductions in both peripheral blood MAIT cell frequencies and MAIT cell functional capacity have been observed in a number of clinical studies of viral infections, including HIV[27] and HTLV-1[26]. However, such studies were correlative, have largely been limited to peripheral cells, and have not addressed local activation and function in the lungs. Most importantly, it remains to be shown in vivo whether activation of MAIT cells plays any role—protective or detrimental—during viral infections[28].

Here, we show that MAIT cells accumulate in the lungs and are activated during in vivo infection with IAV. MAIT cell activation is modulated via cytokines, independently of MR1. Moreover, animals deficient in MAIT cells show exacerbated disease following IAV challenge, and transfer of MAIT cells to immunodeficient mice ameliorates the severity of IAV-induced disease. Together, these data indicate that MAIT cells are activated and contribute to protection against a lethal viral challenge in vivo.

## Results

### MAIT cells accumulate and are activated during influenza.
First, we sought to determine whether MAIT cells were activated in vivo during influenza virus infection. For these studies, C57BL/6 (wild-type; WT) mice received an experimental challenge with 100 plaque forming units (PFU) of the mouse-adapted influenza virus strain A/Puerto Rico/8/34/1934 (PR8, H1N1), which causes severe pneumonia in mice, characterised by parenchymal necrosis and infiltrates of macrophages, lymphocytes and neutrophils (Supplementary Fig. 1). We observed rapid accumulation of pulmonary MAIT cells (defined as TCRβ+CD45.2+CD19− MR1-5-OP-RU tetramer+ cells, see Supplementary Fig. 2), which peaked around 5 days post-infection (dpi) (Fig. 1a, mean 2.6-fold increase from baseline, Kruskal–Wallis with post hoc Dunn's P < 0.0001, n = 20, returning to baseline by day 7 (Fig. 1a, b)). By comparison, peak frequencies of non-MAIT ('conventional') CD8+ cells occurred later at day 7 post-infection. The proportional increase in MAIT cells was significantly higher than that of CD4+ or CD8+ cells early during infection at day 5 post-infection (P < 0.001), though not by day 7 at which point antigen-specific responses will become dominant over other innate-like cells. In addition to MAIT cell recruitment, IAV infection induced MAIT cell activation, as indicated by increased expression of CD25, CD69 and Granzyme B (Fig. 1c–h). Expression of CD25 and CD69 peaked at day 5 post-infection and was significantly higher in MAIT cells than conventional CD4+ or CD8+ T cells, persisting until day 13 post-infection. By contrast, CD25 expression on conventional CD4+ T cells peaked at 7 dpi, whilst CD69 continued to increase over the course of the experiment and the highest levels were detected on CD4+ and CD8+ T cells at 13 dpi. Together, these results indicate that MAIT cells accumulate and are activated early in the airways following IAV infection of mice.

### Accumulation and activation is modulated via cytokines.
We have previously demonstrated that MAIT cells can be activated by viruses independently of MR1, via IL-18 in synergy with IL-12, IL-15 or IFN-α/β[20,21]. We therefore investigated the role of these cytokines in the accumulation and activation of MAIT cells in the lungs following IAV infection, using mouse strains deficient in particular cytokines or cytokine receptors. As an additional control for MR1-mediated activation, intravenous injection was used to adoptively transfer C57BL/6 pulmonary MAIT cells into MR1−/− mice (which otherwise lack MAIT cells), as previously described[14]. Compared with WT C57BL/6 mice, the magnitude of influenza virus-induced pulmonary MAIT cell accumulation was unaffected by deficiency in IL-12, IL-15, IFN-α/β receptor (IFNαR) or MR1 (Fig. 2a, mean 2.1–2.9-fold increases above baseline at 5 dpi), but was significantly impaired (Kruskal–Wallis with post hoc Dunn's P < 0.05) by deficiency in IL-18 (mean 1.3-fold increase) (Fig. 2a).

By contrast, MAIT cell activation (as determined by measurement of CD25 expression) was significantly impaired by deficiency of IL-15, -18 or IFNαR (showing 69%, 70% and 67% reductions in the proportion of MAIT cells expressing CD25 respectively, P < 0.05 for each) and most dramatically by deficiency of IL-12 (90% decrease, P < 0.0001) (Fig. 2b). Differences in CD69 expression were not statistically significant (Fig. 2c). These results support the concept that in vivo, MR1-independent MAIT cell activation is driven by virus-induced pro-inflammatory cytokines, with a dominant role for IL-18, as previously suggested by in vitro studies[21].

### MAIT cell-deficient mice show enhanced mortality.
Given that MAIT cells are activated and accumulate in the lungs following influenza virus infection, we sought to determine whether they might play a protective role or, conversely, enhance the inflammatory cytokine milieu and contribute to exacerbated immune

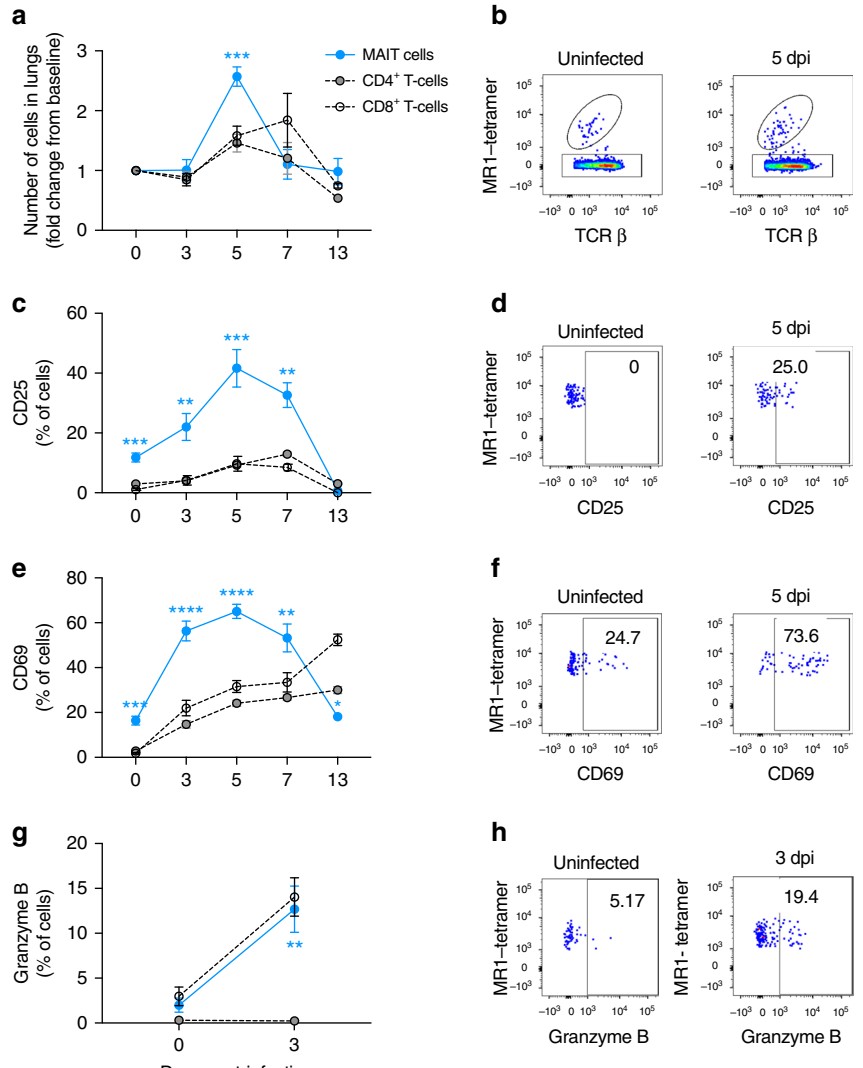

**Fig. 1** Early pulmonary MAIT cell accumulation and activation during influenza virus infection. Accumulation and activation of MAIT cells and MR1-5-OP-RU tetramer-negative 'conventional' CD4+ and CD8+ T cells in C57BL/6 mice before and after challenge with 100 plaque-forming units (PFU) PR8 virus. **a** Accumulation of pulmonary MAIT cells, CD4+ and CD8+ T cells relative to uninfected day 0 controls at 0, 3, 5, 7, 13 days post-infection (dpi) expressed as fold change from baseline, estimated using calibration particles. **b** Representative flow cytometry plots showing TCRβ and MR1-5-OP-RU tetramer staining, gated on live, pulmonary CD45+19− lymphocytes. Proportion of pulmonary MAIT cells expressing **c** CD25, **e** CD69 and **g** Granzyme B expressed as a percentage of MAIT, CD4+ or CD8+ T cells. Representative flow cytometry plots show expression of **d** CD25, **f** CD69 and **h** Granzyme B, gated on live pulmonary CD45+19−TCRβ+ MR1-5-OP-RU tetramer+ lymphocytes. Graphs show combined data (mean ± SEM) from four (days 0, 3, 5), two (day 7) or one (day 13) independent experiments with similar results, with 2–5 mice per group in each replicate. Statistical tests compare MAIT cell subsets with CD4+ and CD8+ subsets at each time-point by Kruskal–Wallis one-way ANOVA with post hoc Dunn's tests; *P < 0.05, **P < 0.01, ***P < 0.001, ****P < 0.0001

pathology. We therefore examined weight loss and survival following influenza virus infection of WT C57BL/6 mice and MR1$^{-/-}$ mice, which have an absolute deficiency of MAIT cells[3,6,29]. Following challenge with 100 PFU of PR8 virus, MR1$^{-/-}$ mice exhibited greater body weight loss (Fig. 3a) and greater mortality (log-rank Mantel–Cox P < 0.0001) (Fig. 3b) in comparison with WT mice. Importantly, the body weight loss and mortality in MR1$^{-/-}$ mice were ameliorated by i.v. adoptive transfer of pulmonary MAIT cells at 1 week prior to influenza virus infection (Fig. 3a, b).

We did not observe significant differences in viral load in lung homogenates prepared from PR8-infected C57BL/6 and MR1$^{-/-}$ mice at days 3 and 5 post-infection (Fig. 3c). However, MR1$^{-/-}$ mice had reduced numbers of SigF+CD11b$^{int}$CD64+CD11c+ alveolar macrophages at 3 dpi (Fig. 3d, Kruskal–Wallis with post

hoc Dunn's P < 0.05) and reduced pulmonary T cell numbers by 5 dpi (Fig. 3e, P < 0.001). The absolute numbers of neutrophils, eosinophils, dendritic cells, NK cells and γδ T cells across the time points post PR8 infection were not statistically significant (Supplementary Fig. 3). Using tetramers for MR1 or to immunodominant epitopes from the nucleoprotein or polymerase acidic protein of influenza virus, we tracked MAIT cells and conventional antigen-specific αβ T cell accumulation during PR8 infection. In MR1$^{-/-}$ mice, we observed a tendency towards decreased accumulation of antigen-specific T cells, at day 9 in splenocytes, and for nucleoprotein at day 7 in the lymph nodes (Supplementary Fig. 4) suggesting MAIT cells may enhance the development of an effective antigen-specific adaptive immune response. Total protein concentrations in cell-free BAL supernatants, a sensitive marker of pulmonary epithelial damage[30],

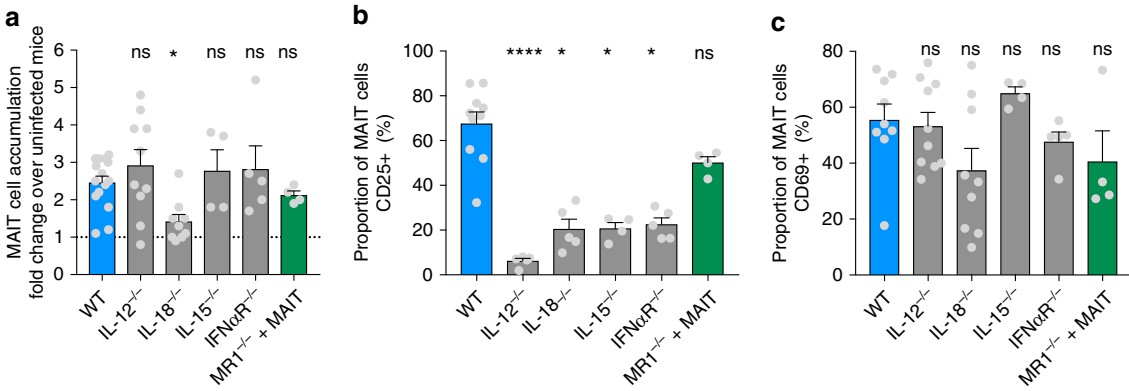

**Fig. 2** MAIT cell accumulation and activation are modulated through cytokines, independent of MR1. Accumulation and activation of MAIT cells in wild type (WT) C57BL/6 mice or mice deficient in IL-12, IL18, IL-15, IFNαR or MR1 5 days after challenge with 100 plaque-forming units (PFU) PR8 virus. Mice deficient in MR1 had received adoptive transfer of $1.8 \times 10^5$ pulmonary MAIT cells 2 weeks earlier, from WT mice previously infected with $10^6$ CFU S. Typhimurium BRD509 for 7 days to expand the MAIT cell population. **a** Fold accumulation of pulmonary MAIT cells relative to uninfected controls. **b**, **c** Proportion of pulmonary MAIT cells expressing CD25 (**b**), and **c** CD69 expressed as a percentage. Graphs show combined data (mean ± SEM) from one (IL-15$^{-/-}$, IFNαR$^{-/-}$, MR1$^{-/-}$) or two (IL-12$^{-/-}$, IL-18$^{-/-}$) independent experiments with similar results. Groups compared with WT by Kruskal–Wallis with post hoc Dunn's tests; $*P < 0.05$, $**P < 0.01$, $****P < 0.0001$

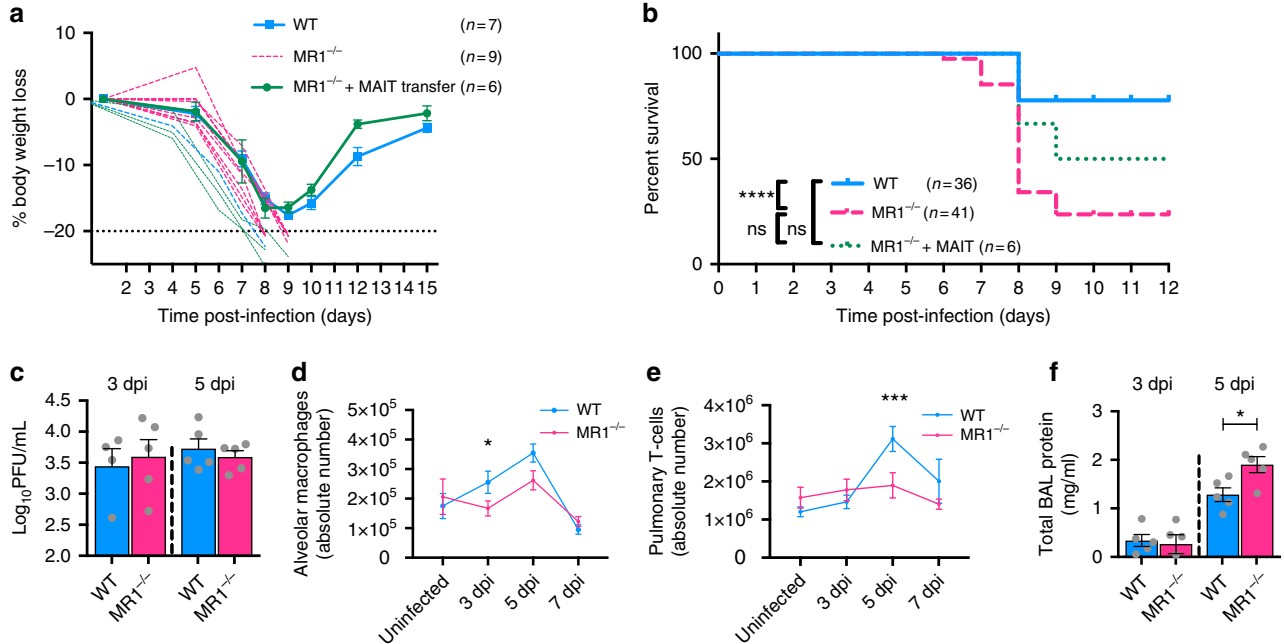

**Fig. 3** MR1$^{-/-}$ mice show enhanced weight loss and mortality in response to severe influenza. **a** Body weight loss expressed as a percentage after infection with 100 PFU PR8 virus. WT ($n = 7$) and MR1$^{-/-}$ data ($n = 9$) are representative of four experiments with similar results. Adoptive transfer ($n = 6$), performed once, used $1 \times 10^6$ pulmonary MAIT cells from WT mice previously infected with $10^6$ CFU S. Typhimurium BRD509 for 7 days to expand the MAIT cell population. Cells were transferred 1 week prior to influenza virus infection. Graphs show mean weights ± SEM for surviving mice, with individual plots for animals which succumbed to infection. **b** Survival curves after intranasal infection with 100 PFU of PR8, showing combined data from one (MR1$^{-/-}$ + MAIT cells, $n = 6$) to four experiments (WT, $n = 36$; MR1$^{-/-}$, $n = 41$). Survival curves compared using log-rank (Mantel–Cox) tests. **c** Titres of infectious virus in clarified lung homogenates at 3 and 5 dpi, expressed as plaque-forming units (PFU). Data are from a single experiment. **d** Absolute numbers of alveolar macrophages and **e** MR1-5-OP-RU tetramer-negative T cells in single cell lung suspensions prepared from uninfected animals and from animals at 3, 5 and 7 dpi. Data are combined from three experiments with similar results. **f** Total protein concentration in cell-free BAL supernatants at days 3 and 5 post-infection. Graphs show means ± SEM. *Mann–Whitney $P < 0.05$; ***$P < 0.001$

were also increased in MR1$^{-/-}$ mice at 5 dpi (Mann–Whitney $P < 0.05$, Fig. 3f). These data indicate that early accumulation of MAIT cells at the site of infection protects against influenza-induced morbidity and mortality associated with excessive lung damage and recruitment of inflammatory cells, but not with viral killing.

As the mouse-adapted PR8 strain is highly virulent in mice, we performed additional experiments using X-31 (H3N2, a

reassortant virus bearing the hemagglutinin and neuraminidase genes from A/Aichi/2/1968 and the remaining six genes from PR8), a strain that shows moderate virulence in mice[30]. After intranasal infection of WT or MR1$^{-/-}$ mice with 5000 PFU of X-31, we did not observe significant differences in survival or in titres of infectious virus detected in the lungs (Supplementary Fig. 5a–c), although we did observe a similar virus-induced pulmonary accumulation of MAIT cells as well as virus-induced

MAIT cell activation (2.6-fold increase at 5 dpi, Supplementary Fig. 5d–g). Furthermore, following infection with either PR8 or X-31, we observed reductions in key innate inflammatory cell cytokines in the lungs of MR1$^{-/-}$ compared with wild-type C57BL/6 mice. In particular, in MR1$^{-/-}$ mice we detected significantly lower amounts of monocyte chemotactic protein 1 (MCP-1/CCL2) and IL-6 following infection with 100 PFU of PR8, and of RANTES/CCL5 following 5000 PFU of X-31 with similar trends observed with both virus strains for each of the inflammatory mediators detected (Supplementary Fig. 6). Likewise, there were lower levels of interferon-γ in bronchoalveolar lavage fluid at day 5 post-infection with 100 PFU of PR8 in MR1$^{-/-}$ mice (3.2-fold difference in geometric mean, $t$-test on log-transformed data, $P = 0.044$) (Supplementary Fig. 7).

Together, these data indicate that MAIT cells play a protective role during severe influenza virus infection, enhancing pulmonary T cell accumulation and innate inflammatory cytokine production, and reducing weight loss, mortality and pulmonary epithelial damage.

**MAIT cell transfer delays mortality in immunodeficient mice.** To further explore the protective role of MAIT cells during influenza infection, we studied the impact of adoptively-transferred MAIT cells into immunodeficient Rag2$^{-/-}$γC$^{-/-}$ mice lacking T cell-, B cell- and NK cell-mediated immunity. As previously described[14,29], ~0.3 × 10$^6$ pulmonary MAIT cells were adoptively transferred into Rag2$^{-/-}$γC$^{-/-}$ mice. Low numbers of residual contaminating conventional T cells from the donor mice can rapidly expand in these recipient mice. Therefore, residual contaminating conventional T cells were depleted in recipients with anti-CD4 and anti-CD8 mAbs, and the animals rested for 1–2 weeks to allow expansion of the MAIT cell population prior to challenge with PR8 or X-31 strains (Fig. 4a). Adoptive transfer of MAIT cells reduced weight loss (Fig. 4b) and significantly prolonged survival after PR8 infection (Fig. 4c, log-rank Mantel–Cox $P < 0.05$). Furthermore, when we directly compared the protective effect of MAIT, CD8 and NK cells in the same model, MAIT cell transfer was associated with prolonged survival compared with CD8 transfer (Bonferroni-corrected log-rank Mantel–Cox $P = 0.009$), but was not superior to NK cell transfer ($P = 0.8$, Supplementary Fig. 8). Moreover, in this immunocompromised Rag2$^{-/-}$γC$^{-/-}$ mouse model adoptive transfer of MAIT cells now also provided protection following challenge with the less virulent X-31 strain (Fig. 4d, e, log-rank Mantel–Cox $P = 0.02$).

Human MAIT cells express IFN-γ in response to stimulation of TLR-8 with single-stranded RNA[20], and also that human viral activation of MAIT cells can suppress replication of hepatitis C virus via IFN-γ in vitro[21], we hypothesised that IFN-γ expression by MAIT cells could contribute to their protective role following influenza virus infection. To address whether IFN-γ expression by MAIT cells confers protection against IAV in vivo, MAIT cells from wild-type C57BL/6 and from IFN-γ$^{-/-}$ donor mice were transferred into Rag2$^{-/-}$γC$^{-/-}$ mice. As expected, the protective effect of adoptively-transferred MAIT cells was abrogated when the transferred MAIT cells were deficient in IFN-γ (Fig. 4d, e), suggesting that the antiviral effects provided by MAIT cells were mediated, at least in part, by IFN-γ.

## Discussion
We have shown in vivo that MAIT cells accumulate in the lungs, are activated and contribute to protection against morbidity and mortality during influenza virus infection in mice. These findings are consistent with our previous human in vitro observations in which we demonstrated MR1-independent MAIT cell activation

in response to exogenous cytokines[20,21] or to IAV[21]. In those experiments, we co-incubated peripheral blood mononuclear cells (PBMCs) or human CD8$^+$ T cells with virus-infected monocytes, macrophages or dendritic cells and observed strong, dose-dependent MAIT cell activation with upregulation of CD69 and IFN-γ and Granzyme B[21]. We also observed virus-induced upregulation of CD69, Granzyme B and IFN-γ on MAIT cells in separate experiments using co-culture of PBMC with IAV infection of a human lung epithelial cell line (A549)[22]. Here, we observed strong CD25 and CD69 upregulation by MAIT cells in the lungs of IAV-infected mice, which occurred earlier and to a greater extent than in non-MAIT CD3+ T cells, and was associated with Granzyme B upregulation, suggesting that these cells are licensed for cytotoxicity early in influenza virus infection.

In the previous in vitro studies, virus-induced responses were unaffected by anti-MR1 blocking antibody[21,22], but rather were driven by cytokines. Indeed, this would be consistent with the MAIT cell accumulation (Fig. 2a) and their protective effects observed after adoptively transferring MAIT cells into mice with deletion of MR1 (Fig. 3b). Stimulation with IL-12 and IL-18 is known to induce IFN-γ from CD161$^{++}$ CD8$^+$ T cells, of which MAIT cells are the dominant subset in humans[20]. In both the lung epithelial cell[22] and the professional antigen presenting cell[21] co-culture systems, we found IL-18 to be the dominant activating cytokine in influenza virus infection, acting synergistically with IL-12 or IL-15 or type I interferons[21]. Again, we confirmed these observations in vivo using mice with genetic deletions of these cytokines or their receptors, and found that MAIT cell accumulation and CD69 upregulation were significantly impaired only in the IL-18$^{-/-}$ mice (Fig. 2a, c). Interestingly, CD25 activation was affected to the greatest extent by IL-12 deletion (Fig. 2b), likely because an important general biological function of IL-12 is to induce T cell upregulation of CD25[31].

Our previous clinical human influenza infection studies showed activation of peripheral blood MAIT cells, measured by Granzyme B upregulation[21] and a decrease in peripheral blood MAIT cell frequencies, which correlated with disease severity[21] and with fatal outcomes[22]. In these studies, it was not possible to determine causal associations nor examine lung tissue. Our murine data recapitulated these findings and revealed activated MAIT cells at the site of infection. In human studies, there are wide inter-individual variations in baseline frequencies of peripheral blood MAIT cells[4]. However, overall the observed clinical association between reduced MAIT cell frequencies and influenza mortality would be consistent with the increased mortality we observed here in MR1$^{-/-}$ mice.

MAIT cells are not unique in their capacity to respond to virus-induced cytokines. Virus-induced bystander activation was first described in conventional CD8$^+$ T cells in mice[32] but is overall a rare event in these cells[33]. However, the high IL-18R expression and IL-12/18 responsiveness appear to be common to a range of innate-like lymphocytes expressing PLZF[34], which in humans include iNKT, γδ T, and NK cells[35]. Indeed, protective roles for iNKT cells have been shown in murine IAV infection[36–38]. Nonetheless, whilst iNKT cells are relatively abundant in mice, they are quite rare in humans, constituting approximately 0.1% of peripheral blood T cells[3,35,39]. In contrast, human MAIT cells are much more abundant comprising 1–10% of blood[3] and pulmonary[4] T cells. Furthermore, at early stages the absolute numbers of these cells in the lungs will markedly exceed the numbers of conventional antigen-specific T cells responding to cognate viral antigens[34]. Furthermore, as they display a significantly higher TCR-independent upregulation of IFN-γ than conventional T cells[22], MAIT cells may have a considerable role in early antiviral protection in humans. Nevertheless, despite much lower relative frequencies of MAIT cells in mice compared

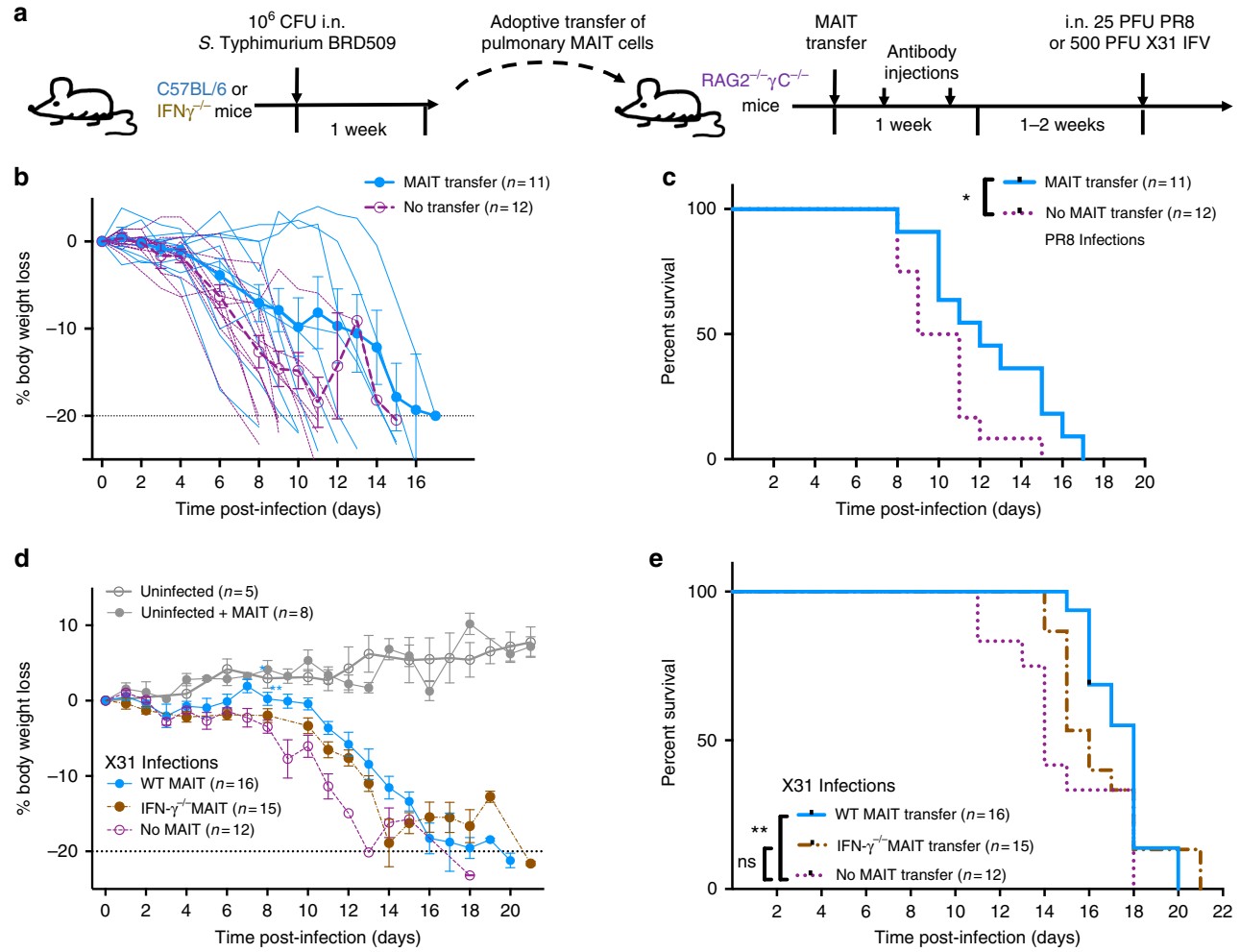

**Fig. 4** Adoptive transfer of MAIT cells delays mortality in Rag2$^{-/-}$γC$^{-/-}$ mice with IAV infection. **a** Schematic of protocol: $3 \times 10^5$ pulmonary MAIT cells from C57BL/6 mice (previously infected with $10^6$ CFU S. Typhimurium BRD509 for 7 days to expand the MAIT cell population) were sorted and transferred intravenously into Rag2$^{-/-}$γC$^{-/-}$ mice, followed by intraperitoneal anti-CD4 and anti-CD8 antibody injection (0.1 mg each) twice within 1 week to deplete any residual conventional T cells included in the transfer. After 2 weeks, mice were infected i.n. with 25 PFU of PR8 (**b+c**) or 500 PFU of X-31 (**d+e**). **b** Body weight loss expressed as a percentage (showing mean ± SEM and individual values for all mice), and **c** survival after infection with 25 PFU PR8 virus. Survival curves compared using log-rank (Mantel–Cox) tests. **d** Body weight loss expressed as a percentage (mean ± SEM), after infection with 500 PFU X-31 virus. *$P < 0.05$, **$P < 0.01$ denotes significant differences between weights using ANOVA and post hoc Dunnett's comparing against non-transferred Rag2$^{-/-}$γC$^{-/-}$ mice. **e** Survival curves after infection with 500 PFU of X-31 were compared using log-rank (Mantel–Cox) tests ($P = 0.02$) and Gehan–Breslow–Wilcoxon tests ($P = 0.006$) between WT MAIT and non-transferred Rag2$^{-/-}$γC$^{-/-}$ mice. Other differences were not significant. Data represent combined data from two (**b**, **c**) or three (**d**, **e**) experiments with similar results

with humans, it is striking that we could observe a protective effect of MAIT cells.

Although MAIT cells protected against influenza virus-induced weight loss and mortality, we did not detect a significant difference in viral titres at the time of peak viral load (i.e. 3–5 dpi). This is not unexpected, as other investigators have also observed significant differences in immunopathology without changes in IAV load[38]. Furthermore, whilst the cytokine responsiveness of MAIT cells suggests the potential for a possible deleterious role in which they could contribute to the 'cytokine storm' effect observed in severe influenza[40], our data from both C57BL/6 and RAG2$^{-/-}$γC$^{-/-}$ strains showed no evidence that MAIT cells contributed to enhanced immunopathology. In fact, the opposite effect was seen. This is a critical observation because, although it is known that MAIT cells home to sites of inflammation[13], are activated by IL-18/-15/-12 to produce IFN-γ[20], and that IFN-γ can be antiviral[41]. Only a direct in vivo challenge experiment is able to directly answer the question as to whether, in a complex

host–pathogen interaction within an intact organism, the protective effects of these mechanisms outweigh the detrimental immunopathological consequences.

The importance of MAIT cells to immune protection in humans may be most apparent when MAIT cell frequencies are reduced. MAIT cell frequencies are lowest in early life[3] and in the elderly[42], age groups at the highest risk of influenza mortality[43]. Influenza mortality is also associated with suppressed cell-mediated immunity, including HIV infection and iatrogenic immunosuppression, conditions in which MAIT cells are also suppressed[43,44]. Of particular relevance to respiratory viral infections, therapeutic use of inhaled and oral corticosteroids causes marked reductions in airway MAIT cell frequencies in asthma[4] and chronic obstructive pulmonary disease[5], and thus may contribute to the increased risk of influenza hospitalisation in these conditions[45].

Influenza virus is a major human respiratory pathogen, which causes seasonal epidemics resulting in 300–650,000 deaths

annually[46]. IAV can also cause pandemic spread when novel strains emerge[43]. Although the most effective method for prevention and control of influenza is vaccination, current vaccines show limited efficacy and little heterologous protection between strains, and so require annual reformulation. The TCR-independent nature of the antiviral MAIT cell response therefore offers significant therapeutic potential in at least two ways. Firstly, strategies to target MAIT cell activation, for instance by incorporating MAIT cell ligands and TLR agonists in the formulation of novel adjuvants, could be used to enhance the efficacy of future influenza vaccines. Secondly, we have shown that MAIT cell frequencies can be rapidly 'boosted' through mucosal administration of simple synthetic MAIT cell ligands with TLR agonists[14,29], which may prove effective in a number of scenarios including short-term prophylactic vaccination during an epidemic. Such a strategy may also prove beneficial in protecting groups known to show impairments in MAIT cell immunity—such as the elderly or those receiving therapeutic corticosteroids—or even acutely during the early stages of infection.

In summary, we have shown for the first time in vivo evidence of an important contribution of MAIT cells to protective immunity against a major human viral pathogen. As the cytokine-mediated activation of MAIT cells is a mechanism likely to be common to a number of other acute and chronic viral infections, as has already been shown in dengue virus, hepatitis C virus in clinical studies[21,28], these findings from an influenza model are likely to be widely applicable to a range of respiratory and systemic viral diseases.

## Methods

**Animal models**. Mice were bred and housed in the Biological Research Facility of the Peter Doherty Institute (Melbourne, Victoria, Australia). MR1$^{-/-}$ mice were generated by breeding Vα19iCα$^{-/-}$MR1$^{-/-}$ mice[47] (from Susan Gilfillan, Washington University, St. Louis School of Medicine, St. Louis, MO) with C57BL/6 mice and inter-crossing of F1 mice. The genotype was determined by tail DNA PCR at the MR1 locus using the following primers[29]: Fwd: AGC TGA AGT CTT TCC AGA TCG; Rev (wild type): ACA GTC ACA CCT GAG TGG TTG; Rev (knockout): GAT TCT GTG AAC CCT TGC TTC. IFN-γ knockout mice (B6.129S7-Ifng$^{tm1Ts}$/J)[48] are commercially available (Jackson Laboratory). Male mice aged 6–12 weeks, matched for age, sex and weight, without randomisation or blinding, were used in experiments, after approval by, and in accordance with, the requirements of the University of Melbourne Animal Ethics Committee (1513661).

**Intranasal virus infection**. Intranasal (i.n.) inoculation with 50–1 × 10$^4$ PFU of X-31 (H3N2; A/Hong Kong/X31) or with 25–150 PFU PR8 (H1N1; A/Puerto Rico/8/34/1934) IAV, or with S. Typhimurium BRD509 (10$^6$ colony forming units (CFU)) in 50 μl per nares was performed on isofluorane-anesthetized mice. Virus stocks were grown in the allantoic cavity of 10 day-old embryonated chicken eggs, and the viral titre was determined by a plaque assay on MDCK monolayers, as previously described[49].

Mice were weighed daily and assessed for visual signs of clinical disease. Animals that lost ≥20% of their original body weight and/or displayed evidence of pneumonia were euthanized.

Mice were killed by CO$_2$ asphyxia, the heart perfused with 10 ml cold RPMI and lungs were taken. To prepare single-cell suspensions, lungs were finely chopped with a scalpel blade and treated with 3 mg ml$^{-1}$ collagenase III (Worthington, Lakewood, NJ), 5 μg ml$^{-1}$ DNAse, and 2% foetal calf serum in RPMI for 90 min at 37 °C with gentle shaking. Cells were then filtered (70 μm) and washed with PBS/2% foetal calf serum. For plaque assays, lungs were placed into RPMI and homogenised using a Polytron System PT 1200 CL 230V (Kinematica, Lucerne, Switzerland). Red blood cells were lysed with hypotonic buffer TAC (Tris-based amino chloride) for 5 min at 37 °C. Approximately 1.5 × 10$^6$ cells were filtered (40 μm) and used for flow cytometric analysis. Absolute cell counts were derived by adding to each sample 2.5 × 10$^4$ blank calibration particles (BD Pharmingen).

**Determination of viral load counts in infected lungs**. Viral load was determined by counting PFU in MDCK monolayers infected with lung homogenates, at varying dilutions for 45 min at 37 °C, 5% CO$_2$ before the addition of an Agarose/L15 or MEM overlay containing Trypsin (Worthington Biochemical, NJ, USA), as described[49]. Plates were incubated at 37 °C, 5% CO$_2$ for 3 days before plaques were counted.

**Adoptive transfer**. As MAIT cell numbers are low in naive C57BL/6 mice, prior to adoptive transfer experiments MAIT cell populations were expanded by intranasal infection with 10$^6$ CFU S. Typhimurium BRD509 in 50 μl PBS for 7 days, as described[29]. After 7 days, mice were sacrificed, single-cell suspensions prepared and live CD3+CD45+MR1-5-OP-RU tetramer+ cells sorted using a BD FACS Aria III. Simultaneously, from these single cell suspensions, live CD3+CD45+CD8+ MR1-5-OP-RU tetramer− were sorted for CD8+ T cell adoptive transfer. For the transfer of NK cells, prior to cell sorting, single cell suspensions from naive WT spleens were subjected to magnetic bead-based antibody depletion with anti-CD11b, anti-CD4, anti-CD8 and anti-B220 (reagents kindly provided by Professor Axel Kallies). Live NK1.1+CD3-CD4-CD8-B220-CD11b-CD11c- cells were sorted using a BD FACS Aria III. 3 × 10$^5$ pulmonary MAIT cells were injected into the tail veins of recipient Rag2$^{-/-}$γC$^{-/-}$ mice which then received 0.1 mg each of anti-CD4 (GK.5) and anti-CD8 (53.762) mAb i.p. on days 2 and 5 or 6 to control residual donor-derived conventional T cells. Mice were rested for 2 weeks post transfer to allow full expansion of the MAIT cell population prior to infectious challenge. In a separate set of experiments, 1 × 10$^5$ pulmonary MAIT or CD8+ cells or splenic NK cells were transferred to recipient Rag2$^{-/-}$γC$^{-/-}$ mice and rested as described above. Prior to infectious challenge, venous tail blood was stained to confirm the presence of the transferred cell subsets.

**Reagents**. MR1-5-OP-RU-SA BV421 tetramers were generated as described previously[9] (now available from the NIH core tetramer facility) and influenza-specific MHC Class I tetramers targeting immunodominant epitopes of nucleoprotein (D$^b$NP$_{366}$) and polymerase acidic protein (D$^b$PA$_{224}$) and were utilised conjugated to SA-APC and PE respectively. Antibodies against murine CD3e (clone 145-2C11, catalogue number 561108, fluorochrome PerCP-Cy5.5, dilution 1:200), CD4 (GK1.5, 552051, APC-Cy7, 1:200), CD11b (M1/70, 563015, BV605, 1:600), CD19 (1D3, 561113, PerCP-Cy5.5), CD45.2 (104, 553772, FITC and 563685, BV711, 1:400), CD69 (H1.2F3, 12-0691-81, PE and 70318, FITC), IFN-γ (XMG1.2, 557649, PE-Cy7), Ly6C (AL-21, 561237, AF700, 1:400), Ly6G (IA8, 560601, PE-Cy7, 1:1000), NK1.1 (PK136, 553165, PE, 1:200), SiglecF (E50-2440, 562757, PECF594, 1:1000), TCRβ (H57-597, 553174, APC or 553172, PE, 1:500) were purchased from BD (Franklin Lakes, NJ). Antibodies against CD3 (17A2, 56-0032-80, AF700, 1:100) CD8α (53-6.7, 12-0081-81, PE), CD11c (N418, 17-01114-81, FITC, 1:1000), CD25 (PC61.5, 17-02510-81, APC), F4/80 (BM8, 11-4801-81, FITC, 1:100), Granzyme B (NGZB, 12-8898-80, PE) were purchased from eBioscience (San Diego, CA). Abs against CD19 (6D5, 115546, BV510), F4/80 (BM8, 123116, APC), CD11b (M1/70, 101206, FITC, 1:1000), CD11c (N418, 117336, BV786, 1:400), CD19 (6D5, 115506, FITC, 1:500), CD27 (LG.3A10, 124214, PerCP-Cy5.5, 1:400), CD62L (Mel-14, 104406, FITC, 1:800 and PE-Cy7, 1:500), CD64 (X54-5/71, 139322, AF647, 139311, BV711, 1:400, I-Ab, AF6-120.1, 116422, Pacific Blue, 1:400) were purchased from Biolegend (San Diego, CA). To block non-specific staining, cells were incubated with unlabelled MR1-6-formylpterin tetramer and anti-Fc receptor (2.4G2) for 15 min at room temperature and then incubated at room temperature with Ab/tetramer cocktails in PBS/2% foetal calf serum. 7-aminoactinomycin D (5 μl per sample) was added for the last 10 min or fixable viability stains added according to the manufacturer's instructions: Live/Dead Fixable Aqua (L34957, ThermoFisher, Waltham, MA, 1:800) and ZombieYellow (423104, Biolegend).

Cells were fixed with 1% paraformaldehyde prior to analysis on LSRII or LSR Fortessa or Canto II (BD Biosciences) flow cytometers. For intracellular staining, GolgiPlug (BD Biosciences) was used during all processing steps. Surface staining was performed at 37 °C, and cells were stained for intracellular molecules using the BD Fixation/Permeabilization Kit (BD, Franklin Lakes, NJ) or transcription factors using the transcription buffer staining set (eBioscience) according to the manufacturer's instructions.

The concentration of protein in BAL supernatants was measured by adding Bradford protein dye (BioRad) to diluted BAL supernatant according to the manufacturer's instructions. A standard curve using bovine serum albumin was constructed, and the optical density (OD) was determined at 595 nm.

**Cytometric bead analysis**. The CBA flex set (BD Bioscience) was used as per manufacturer's instructions to determine the cytokine concentrations of Lung and BAL supernatant[49]. Data were acquired using FACSCantoII and analysis was by FCAP Array software (Soft Flow Inc., Pecs, Hungary).

**Statistical analysis**. Statistical tests were performed using the Prism GraphPad software (version 7.0, La Jolla, CA). Comparisons were performed using Mann–Whitney tests (two groups) with Bonferroni correction for multiple comparisons, or with Kruskal–Wallis one-way ANOVA with post hoc Dunn's tests (multiple groups). Survival curves were compared using log-rank (Mantel–Cox) tests (two groups) or the Gehan–Breslow–Wilcoxon method for multiple groups. Variance was similar in each case between groups being compared. Sample sizes were estimated based on previous extensive experience in the laboratory with these PR8 and X-31 strains. All statistical tests are two-sided. Flow cytometric analysis was performed with FlowJo10 software (Ashland, OR).

**Reporting Summary**. Further information on research design is available in the Nature Research Reporting Summary linked to this article.

## Data availability
Data supporting the findings of this study are available within the paper and its supplementary information files, or are available from the authors upon request.

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

## Acknowledgements
B.v.W. was supported by the Royal Society (IE160540). The research leading to these results has received funding from the People Programme (Marie Curie Actions) of the European Union's Seventh Framework Programme (FP7/2007-2013) under REA grant agreement number 608765. The content represents only the authors' views and not those of the European Commission. T.S.C.H. is supported by a Wellcome Trust Postdoctoral Research Fellowship (104553/z/14/z). The work was supported by the National Health and Medical Research Council of Australia (NHMRC) Program Grants 1113293, 1071916, 1016629 and 606788, and Project Grant 1120467. A.J.C. is supported by an

ARC Future Fellowship. S.B.G.E. is supported by an ARC DECRA Fellowship. H.W. is supported by a Melbourne International Engagement Award (University of Melbourne). C.D'S. is supported by a Melbourne International Research Scholarship and a Melbourne International Fee Remission Scholarship (University of Melbourne). S.S. is a recipient of Victoria India Doctoral Scholarship and Melbourne International Fee Remission Scholarship, University of Melbourne. M.K. and S.N. are recipients of Melbourne International Research Scholarship and Melbourne International Fee Remission Scholarships. D. I.G. is supported by an NHMRC Senior Principal Research Fellowship (1117766) and by the Australian Research Council (ARC; CE140100011). K.K. is an NHMRC Senior Research Level B Fellow. P.K. was supported by an NIHR Senior Fellowship, Oxford Martin School (PK) and the Wellcome Trust (WT109965MA). A.K. was supported by a Sylvia and Charles Viertel fellowship. We are grateful to the Doherty Institute Flow Cytometry Facility and to Prof. Ian van Driel, Dr. Sammy Bedoui, Dr. Thomas Gebhardt and Dr. Julia Prier, for their kind provision of $IL12^{-/-}$, $IL15^{-/-}$, $IL18^{-/-}$ and $IFN\alpha R^{-/-}$ mice, reagents, intellectual and technical expertise.

## Author contributions

B.v.W., L.L., T.S.C.H., Z.C., T.J.P., H.W., M.S., Z.Z., M.K., S.N., S.S., Z.W., X.J., C.D'S., C. F.A., L.K., S.B.G.E., B.S.M., performed the experiments. B.v.W., L.L., T.S.C.H. analyzed the data. B.v.W., T.S.C.H., L.L., J.M., D.I.G., A.K., P.C.R., A.J.C., P.K., K.K. designed the experiments and managed the study. B.v.W., T.S.C.H., P.K., J.M. conceived the work and wrote the manuscript which was revised and approved by all authors. These authors contributed equally to the supervision of this work: J.M., P.K., K.K., T.S.C.H.

## Additional information

**Competing interests:** Z.C., S.B.G.E., J.M. and A.J.C. are inventors on patents describing MR1 tetramers. The remaining authors declare no competing interests.

