## [Peer Review File · Nature Communications]

Reviewers' comments:

Reviewer #1 (Remarks to the Author):

In this study Van Wilgenburg and colleagues have investigated the contribution of MAIT cells to protection against influenza virus infection in mice. They observe that MAIT cells are rapidly activated *in vivo* and accumulate in the lungs of mice infected with the PR8 influenza strain. They use knock-out mice to demonstrate that this effect is independent of MR1 and dependent on cytokines IL-18 (homing) and IL-15/12 (activation). Again using the PR8 strain, they go on to observe increased morbidity and mortality on MR1^{-/-} mice as compared to wild type, and this was ameliorated by adoptive transfer of MAIT cells. As the PR8 strain is highly pathogenic in mice the authors then tested the less pathogenic X-31 strain, and there disappointingly the effects of MR1 and MAIT deficiency were not repeated. Yet, on the immunocompromised RAG2^{-/-} commongamma^{-/-} background, adoptively transferred MAIT cells could afford some degree of protection against both strains. Finally, the protective effect was found to at least partly depend on IFN γ . This reviewer has the following concerns:

- 1) The data set is consistent with a detectable but relatively modest effect of MAIT cells in protection from influenza virus infection. The observation of a MAIT cell mediated protective effect against viral infection is novel and important as such. However, all observations in this paper are completely expected as all these events described have been observed previously in different settings *in vitro* or *in vivo*. It is known that MAIT cells home to inflammation, are activated by IL-18/15/12 to produce IFN γ , and that IFN γ has effects on viruses such as influenza. So in a sense it would have been more surprising if MAIT cells had no effect in this murine model system.
- 2) The PR8 strain seems to be used at one dose (100 PFU). It could be very informative to titrate the dose to get a feeling for the magnitude of the protective effect of MAIT cells.
- 3) Likewise, it would be interesting to compare effects of adoptive transfer of MAIT cells, NK cells and CD8 T cells in the RAG2^{-/-}-Commongamma^{-/-} host.
- 4) In figure 1a it is stated that the number (not percentage) of cells were measured. How were the numbers established?

Reviewer #2 (Remarks to the Author):

Comments to Authors:

This report follows up earlier studies by this group on the role of MAIT cells in the host response to infection. In the current report, the authors focus on *in vivo* studies on the role of MAIT cells in recovery from experimental IAV infection in the mouse model. The findings in this report, which are largely based on results employing MR – 1 knockout mice and adoptive transfer of MAIT cells into MR – 1 knockout mice or adaptive immune deficient mice, suggest that MAIT cells play beneficial role in recovery from experimental IAV infection.

Since available evidence suggests that MAIT cells do not directly recognize and lyse virus infected cells, the mechanism by which MAIT cells function in this model is not completely clear. The author suggests that the cells may be operating early in infection by producing interferon gamma (Although direct role for interferon gamma as a critical mediator of recovery from experimental IAV infection is not completely clear from the literature). Accordingly this report would be considerably strengthened if the authors provided data on the production of interferon gamma in the BAL of PR 8 and X – 31 infected MR – 1 knockout mice early in infection i.e. for example days 3, 5 and, 7 after infection.

The authors further suggest that IL – 18 is playing an important role in the accumulation of MAIT cells in the infected lungs. Do IL-18 knockout mice show the same enhanced susceptibility to lethal

infection as MR – 1 knockout mice and if so. Can adoptive transfer of MAIT cells conferring hands resistance in IL-18 knockout mice?

The authors compare the tempo of accumulation and activation of MAIT cells in the infected lungs relative to “conventional” (my quotations) CD4 and CD8 T cells. Since up to day 6 – 7 post infection most of the conventional T cells accumulating in the lungs are not antigen specific, one might argue this is not an appropriate comparison. The author should include data on the kinetics and activation state of antigen specific CD8 and CD4 T cells in the infected lungs. In this connection the author should provide data on the absolute number of MAIT cells and antigen specific CD8 T cells accumulating in the lungs over time following A/PR 8 infection as they do in the supplemental figures for X – 31 infection. The data is particularly important to better understand the mechanism of action of MAIT cells as one possible role of the cells is to enhance the development of an effective antigen specific adaptive immune response which would account for the enhanced resistance to lethal infection in MR – 1 deficient animals.

Reviewer #1 (Remarks to the Author):

In this study Van Wilgenburg and colleagues have investigated the contribution of MAIT cells to protection against influenza virus infection in mice. They observe that MAIT cells are rapidly activated *in vivo* and accumulate in the lungs of mice infected with the PR8 influenza strain. They use knock-out mice to demonstrate that this effect is independent of MR1 and dependent on cytokines IL-18 (homing) and IL-15/12 (activation). Again using the PR8 strain, they go on to observe increased morbidity and mortality on MR1^{-/-} mice as compared to wild type, and this was ameliorated by adoptive transfer of MAIT cells. As the PR8 strain is highly pathogenic in mice the authors then tested the less pathogenic X-31 strain, and there disappointingly the effects of MR1 and MAIT deficiency were not repeated. Yet, on the immunocompromised RAG2^{-/-}-commongamma^{-/-} background, adoptively transferred MAIT cells could afford some degree of protection against both strains. Finally, the protective effect was found to at least partly depend on IFN γ .

This reviewer has the following concerns:

1) The data set is consistent with a detectable but relatively modest effect of MAIT cells in protection from influenza virus infection. The observation of a MAIT cell mediated protective effect against viral infection is novel and important as such. However, all observations in this paper are completely expected as all these events described have been observed previously in different settings *in vitro* or *in vivo*. It is known that MAIT cells home to inflammation, are activated by IL-18/15/12 to produce IFN γ , and that IFN γ has effects on viruses such as influenza. So in a sense it would have been more surprising if MAIT cells had no effect in this murine model system.

We agree with the reviewer that many of the observations in this paper might reasonably be predicted by extrapolation of data from *in vitro* experiments and from cross-sectional, observational human data from peripheral blood, and this is important in lending validity and robustness to our findings which are strongly hypothesis-driven. Nonetheless, as we state in the introduction, it very much remained to be shown *in vivo* whether activation of MAIT cells plays any role - protective or detrimental - during viral infections.

Specifically, it could equally be hypothesised that non-specific, cytokine-mediated activation of MAIT cells could be detrimental by contributing to the immunopathology of influenza virus infection, as there is a well-described phenomenon of morbidity and mortality arising from the 'cytokine storm' which can occur during the early stage of human influenza.

Equally it could also be hypothesised that this cytokine-mediated activation of MAIT cell IFN- γ production might be of no real physiological significance *in vivo*. Although, the data we present in Supplementary Fig 4 show that with a less pathogenic influenza strain (X31), MAIT cell activation is still potently triggered.

Therefore, in light of the reviewer's comment, we have enhanced our discussion of this point in the Discussion section of the manuscript with insertion of the following additional text (page 13):-

"This is a critical observation because, although it is known that MAIT cells home to sites of inflammation, are activated by IL-18/-15/-12 to produce IFN- γ , and that IFN- γ can be antiviral. Only a direct *in vivo* challenge experiment is able to directly answer the question as to whether, in a complex host-pathogen interaction within an

intact organism, the protective effects of these mechanisms outweigh the detrimental immunopathological consequences.”

2) The PR8 strain seems to be used at one dose (100 PFU). It could be very informative to titrate the dose to get a feeling for the magnitude of the protective effect of MAIT cells.

We appreciate the reviewer’s comment, although there are technical challenges with repeating these experiments at different doses, particularly within the constraints provided by our institutional review board who limit acceptable weight loss to a max of 20% initial body weight. We note that other institutes worldwide may accept weight losses of 25%, 30% or even 35%.

We have titrated our stocks of virus over a range of infective doses. As can be seen from the summary data below doses significantly higher than 100 PFU (in this instance 500 PFU) are rapidly fatal.

Figure above: titrations of infective inoculae of PR8 virus stocks performed in C57BL/6 mice on two occasions, with n=3-8 mice/ group. Mice losing >20% weight or which physically appeared moribund were euthanased immediately. Data plotted as summary statistics (mean +/-SEM).

Indeed, in previous work with a previous batch of the same strain of PR8, our team have shown all mice infected with 100 PFU of PR8 lost weight progressively and succumbed to disease within 9-10 d (Tate, Schilter et al. 2011), whilst others have shown that for ‘PR8, 100 PFU are equivalent to 5000 EID₅₀, which is equivalent to 1 LD₅₀.’ (Powell, Strutt et al. 2007), suggesting the dose we have selected for this study is close to the upper limit of what is survivable. Therefore, in view of the prior experience of our group with this strain, and the titration experiments we have performed with our stocks, and the severity limits within our Institute, we do not think that we could generate useful data with higher initial PR8 inoculae: we would expect all mice to succumb.

Conversely, with respect to lower inoculae, the data presented above show that at less than 50 PFU no significant systemic disease or mortality occur. Moreover, when the data comprising this graph are plotted as individual values, it is clear that at or below 50 PFU some mice essentially experience no significant weight loss: i.e. at such low inoculae infection can become an unpredictable stochastic process, making reliable low dose infectious challenges difficult.

Figure above: titrations of infective inoculae of PR8 virus stocks performed in C57Bl/6 mice on two occasions, with n=3-8 mice/ group. Mice losing >20% weight or which physically appeared moribund were euthanased immediately. Data plotted as individual data points.

Consequently, due to these technical limitations, we have not attempted the comparative C57Bl/6 v MR1^{-/-} v MR1^{-/-} and adoptive transfer experiments at other ranges of inoculae.

3) Likewise, it would be interesting to compare effects of adoptive transfer of MAIT cells, NK cells and CD8 T cells in the RAG2^{-/-}-Commongamma^{-/-} host.

This is an interesting suggestion. We have therefore performed the experiments to do this, repeating the experiments previously presented in Figure 4 a-c in which MAIT cells, NK cells and CD8⁺ T cells are adoptively transferred into the Rag2^{-/-}γC^{-/-} mice before PR8 influenza infection. We used 5 mice per group and preformed the experiment twice to confirm reproducibility. Again, we see evidence of MAIT cell protection with the transfer: 1 animal in the MAIT group survived till the end of the experiment, and survival was longest in the MAIT transferred group, although, given the slightly smaller group size this difference did not reach significance compared with NK transfer or no transfer. Nonetheless, survival was significantly longer with MAIT cell transfer than with the transfer of CD8⁺ T cells (CD8+TCRβ+ CD45.2+CD11b-CD11c-B220-F4/80-) alone (P=0.009).

We now include the following additional text in the Methods (pg 16) and Results (page 10):-
Methods, pg 16,

“Simultaneously, from these single cell suspensions, live CD3⁺CD45⁺CD8⁺MR1-5-OP-RU tetramer-negative cells were sorted for CD8⁺ T cell adoptive transfer. For the transfer of NK cells, prior to cell sorting, single cell suspensions from naïve WT spleens were subjected to magnetic bead-based antibody depletion with anti-CD11b, anti-CD4, anti-CD8 and anti-B220 (reagents kindly provided by Professor Axel Kallies). Live NK1.1+CD3-CD4-CD8-B220-CD11b-CD11c- cells sorted using a BD FACS Aria III. 3x10⁵ pulmonary MAIT cells were injected into the tail veins of recipient Rag2^{-/-}γC^{-/-} mice which then received 0.1 mg each of anti-CD4 (GK.5) and anti-CD8 (53.762) mAb i.p on days 2 and 5 or 6 to control residual donor-derived conventional T cells. Mice were rested for 2 weeks post transfer to allow full expansion of the MAIT cell population prior to

infectious challenge. In a separate set of experiments 1×10^5 pulmonary MAIT or $CD8^+$ cells or splenic NK cells were transferred to recipient $Rag2^{-/-}\gamma C^{-/-}$ mice and rested as described above. Prior to infectious challenge, venous tail blood was stained to confirm the presence of the transferred cell subsets.”

Results Pg10,

‘. Furthermore, when we directly compared the protective effect of MAIT, $CD8^+$ and NK cells in the same model MAIT cell transfer was associated with prolonged survival compared with $CD8^+$ T cell transfer (Bonferroni-corrected log-rank Mantel-Cox $P=0.009$), but was not superior to NK cell transfer ($P=0.8$, Supplementary Fig.7). Moreover, in...’

We have included the following supplementary figure (Supplemental Figure 7, Supplement Page 11):-

Supplemental Figure 7. Comparative effect of adoptive transfer of MAIT cell, NK cells and CD8 cells in $Rag2^{-/-}\gamma C^{-/-}$ mice with IAV infection.

1×10^5 pulmonary MAIT ($CD3^+CD45.2^+TCR\beta^+MR1-5-OP-RU$ tetramer $^+$), $CD8^+$ ($TCR\beta^+CD45.2^+CD11b^-CD11c^-B220^+F4/80^+CD8^+$) or splenic NK ($CD45.2^+CD8^+CD11b^-CD11c^-B220^+F4/80^+NK1.1^+$) cells from C57BL/6 mice (MAIT and CD8 were derived from C57BL/6 mice previously infected with 10^6 CFU *S. Typhimurium* BRD509 for 7 days to expand the MAIT cell population) were sorted and transferred intravenously into $Rag2^{-/-}\gamma C^{-/-}$ mice, followed by intraperitoneal anti-CD4 and anti-CD8 antibody injection (0.1 mg each) twice within 1 week to deplete any residual conventional T cells included in the MAIT cell transfer. After 2 weeks, mice were infected i.n. with 25 PFU of PR8. For schematic see Fig 4a. (a) Body weight loss expressed as a percentage (showing mean \pm SEM), and (b) survival curves compared using log-rank (Mantel-Cox) tests ($P=0.004$) and Gehan-Breslow-Wilcoxon tests ($P=0.012$) between groups. * $P<0.05$, ** $P<0.01$ denote significant differences in *post hoc* comparisons between individual groups using log-rank tests with Bonferroni correction. Other differences were not significant. Data represent combined data from two experiments with similar results, each using 5 mice / group, per replicate.

4) In figure 1a it is stated that the number (not percentage) of cells were measured. How were the numbers established?

We are able to measure both relative (percentage) and absolute numbers of cells in our assays, by adding to each sample a known absolute number of counting beads, in this case 25,000 6.0-6.4 μm spherical blank calibration beads per sample. These beads are manually counted under a microscope prior to addition, then detected in the flow cytometer during sample acquisition, enabling a precise estimation of the absolute number of any event using the ratio of the number of calibration beads detected to the number of events of interest detected according to the following formula:

$$\text{Absolute no. of events} = \frac{\text{no. of events of interest detected} \times \text{no. of beads added}}{\text{no. of beads detected} \times \text{proportion of total tissue used for cytometry}}$$

Therefore, we have revised the figure legend to Figure 1a by insertion of the text (page 26):-
“, estimated using calibration particles.”

And we have added the following text to the methods (page 15):-

“Absolute cell counts were derived by adding to each sample 2.5×10^4 blank calibration particles (BD Pharmingen).”

Reviewer #2 (Remarks to the Author):

Comments to Authors:

This report follows up earlier studies by this group on the role of MAIT cells in the host response to infection. In the current report, the authors focus on in vivo studies on the role of MAIT cells in recovery from experimental IAV infection in the mouse model. The findings in this report, which are largely based on results employing MR – 1 knockout mice and adoptive transfer of MAIT cells into MR – 1 knockout mice or adaptive immune deficient mice, suggest that MAIT cells play beneficial role in recovery from experimental IAV infection.

Since available evidence suggests that MAIT cells do not directly recognize and lyse virus infected cells, the mechanism by which MAIT cells function in this model is not completely clear. The author suggests that the cells may be operating early in infection by producing interferon gamma (Although direct role for interferon gamma as a critical mediator of recovery from experimental IAV infection is not completely clear from the literature). Accordingly this report would be considerably strengthened if the authors provided data on the production of interferon gamma in the BAL of PR 8 and X – 31 infected MR – 1 knockout mice early in infection i.e. for example days 3, 5 and, 7 after infection.

As suggested by the reviewer, we have conducted additional assays for interferon- γ in bronchoalveolar lavage fluid which was available from mice on days 5 and 7 after 100 PFU PR8 virus infection (although unfortunately samples were not available on day 3 or after X31). This does indeed show evidence of a trend in the reduction of interferon gamma in MR1^{-/-} mice at day 5 post infection ($p=0.056$), coinciding with the peak of accumulation and activation of MAIT cells. This is therefore consistent with the findings from the separate IFN- γ ^{-/-} adoptive transfer experiments which suggest the MAIT cell protection is mediated, at least in part, by IFN- γ and does strengthen our findings.

We have inserted the following additional text in results (page 9):-

“Likewise we detected a trend towards lower levels of interferon- γ in bronchoalveolar lavage fluid at day 5 post infection with 100 PFU of PR8 in MR1^{-/-} mice (3.2-fold difference in geometric mean, P=0.056) (Supplementary figure 6).”

We have inserted the following additional supplementary figure (Supplementary Figure 6, Supplement page 10):-

Supplemental Figure 6. Interferon- γ in bronchoalveolar lavage fluid from wild-type and MR1^{-/-} mice during influenza virus infection.

Concentrations of interferon- γ measured by cytokine bead array in bronchoalveolar lavage fluid on days 5 and 7 post-infection with 100 PFU of PR8 virus compared in WT (C57BL/6) and MR1^{-/-} mice. Experiment performed once. Mann-Whitney test on log-transformed data. Graphs show means of n=5 mice / group.

In addition, we have conducted additional experiments (presented below and as new supplementary figure 3) to investigate the kinetics of development of antigen-specific T cell responses which provides some evidence that MAIT cells may, at least in part, act to enhance the development of antigen a specific T cell responses. See below, page 9.

The authors further suggest that IL – 18 is playing an important role in the accumulation of MAIT cells in the infected lungs. Do IL-18 knockout mice show the same enhanced susceptibility to lethal infection as MR – 1 knockout mice and if so. Can adoptive transfer of MAIT cells conferring [enhanced] resistance in IL-18 knockout mice?

Interleukin-18 deficient mice do indeed have a markedly enhanced susceptibility to lethal influenza virus, as has been reported by others (Liu, Mori et al. 2004), and as can be seen from our data presented below, in which it can be seen that IL-18^{-/-} mice have more severe disease

and more rapid weight loss than MR1^{-/-} mice, and were considered moribund by day 5, requiring euthanasia.

Figure above: body weight loss after 100 PFU PR8 virus infection in wild type mice (with or without MAIT cell activation by 5OPRU+CpG) compared with mice deficient in MR1, IL-12 and IL-18. Data plotted as summary statistics (mean \pm SEM). IL-18 mice were euthanased at day 5 due to rapid weight loss and rapidly deteriorating clinical condition, appearing moribund.

Unfortunately, because IL-18 is such an essential component of anti-viral defence, acting also on cells other than MAIT cells, which include natural killer cells (Liu, Mori et al. 2004) and Th1 cells (Van Der Sluijs, Van Elden et al. 2005), the experiment suggested would be difficult. Indeed, in the experiment proposed, as MAIT cells require IL-18 for their accumulation and activation in this system, an IL-18^{-/-} mouse would not provide the necessary signal to activate any adoptively transferred wild-type MAIT cells, and we would therefore not expect them to be able to rescue this defect. Moreover, as the IL-18^{-/-} phenotype is so marked in this system, and as MAIT cells are not the sole source of IL-18, it is unlikely that even fully activated MAIT cells would be able to rescue this defect.

A more relevant experiment might be adoptive transfer of MAIT cells into an IL-18R^{-/-} mouse. We have not performed this experiment both for practical reasons (because the mice are not available to us, and we do not have the appropriate ethical approval) and for scientific reasons: as with the IL-18^{-/-} mouse, the IL-18R deficiency would affect the immune system much more broadly than just the MAIT cells, and so the protective effects of MAIT cells would almost certainly be masked by the deficiency in NK and conventional T cell responses. A cre-flox system in which IL-18R deficiency is restricted to MAIT cells (and not other PLZF-expressing cells) is not currently available, however we think the reviewer's suggestion is interesting but not within our capacity to test.

The authors compare the tempo of accumulation and activation of MAIT cells in the infected lungs relative to "conventional" (my quotations) CD4 and CD8 T cells. Since up to day 6 – 7 post infection most of the conventional T cells accumulating in the lungs are not antigen

specific, one might argue this is not an appropriate comparison.

The reviewer's comment that prior to day 6/7 most of the conventional T cells accumulating in the lungs are not antigen-specific is in accord with the new data on the kinetics of antigen-specific T cell recruitment, which we provide in the new Supplementary figure 3 b, in which antigen-specific T cell frequencies are undetectable at day 5, but comparable to or in excess of MAIT cell frequencies by day 7. However, this is exactly the point we are making: that MAIT cells (which, given they possess TCRs specific only to bacterial ligands, are also not antigen-specific in this context) can provide protection via a non-specific, cytokine-mediated accumulation and activation earlier than antigen-specific T cells, during the critical time window in which viruses can replicate unchecked by adaptive immunity. This is shown graphically in Figure 1a and Supplementary figure 3 b. We do, however, accept that there may be other innate-like cells – iNKT, NK and innate lymphoid cells – which may also be active during this early time frame, and we do not intend to claim that MAIT cells are the only source of early innate cytokines.

Therefore, to clarify this point for the reader we have expanded the relevant sentence (results page 6):-

By comparison, peak frequencies of non-MAIT ('conventional') CD8⁺ cells occurred later at day 7 post-infection. The proportional increase in MAIT cells was significantly higher than that of CD4⁺ or CD8⁺ T cells **early during infection** at day 5 post-infection (P<0.001), **though not by day 7 at which point antigen-specific T cell responses will become dominant over other innate-like cells.**

The author should include data on the kinetics and activation state of antigen specific CD8 and CD4 T cells in the infected lungs. In this connection the author should provide data on the absolute number of MAIT cells and antigen specific CD8 T cells accumulating in the lungs over time following A/PR 8 infection as they do in the supplemental figures for X – 31 infection. The data is particularly important to better understand the mechanism of action of MAIT cells as one possible role of the cells is to enhance the development of an effective antigen specific adaptive immune response which would account for the enhanced resistance to lethal infection in MR – 1 deficient animals.

We thank the reviewer for their suggestion and have therefore conducted additional experiments, as requested, to show the relative kinetics of absolute numbers of tetramer-labelled MAIT cells and of conventional $\alpha\beta$ T CD8⁺ T cells recognising two different immunodominant epitopes, D^bNP₃₆₆ and D^bPA₂₂₄ with differential kinetics and derived from the nucleoprotein (NP) and polymerase acidic (PA) protein from influenza A, respectively. We observed a transient increase in MAIT cell frequencies, accompanied by the appearance of virus-specific T cells. We compared wild-type and MR1^{-/-} mice and observed a tendency towards greater antigen-specific T cell responses in MAIT-sufficient wild-type mice, which would be consistent with the reviewer's suggestion that MAIT cells could enhance the development of an effective antigen specific adaptive immune response.

To present these findings we have inserted the following additional supplementary figure below (Supplemental Figure 3, Supplement page 4-5):-

Supplemental Figure 3. Kinetics of tissue accumulation of MAIT cells and antigen-specific conventional CD8⁺ T cells in wild-type and MR1^{-/-} mice with IAV infection.

(a) Representative flow cytometry plots showing TCR β and MR1-5-OP-RU tetramer staining (left), CD8 α and nucleoprotein (NP₃₆₆) tetramer (middle), and CD8 α and polymerase acidic protein (PA₂₂₄) tetramer (right), gated on live, pulmonary CD45⁺CD3⁺B220⁻CD11c⁻F4/80⁻ lymphocytes. (b) Changes in absolute numbers of MAIT cells and of conventional antigen-specific CD8 α ⁺ T cells specific for NP₃₆₆ and PA₂₂₄ in the lungs of wild-type mice 5, 7 and 9 days after intranasal infection with 100 PFU of PR8 virus. (c-i) Absolute numbers of all live cells (c) and of antigen-specific conventional T cells (d-i) in the lungs (c-e), spleen (f,g) and mediastinal lymph nodes (h,i) of wild-type and MR1^{-/-} mice 5, 7 and 9 days after

intranasal infection with 100 PFU of PR8 virus. Experiment performed once in 4-5 (day 5, 7) or 2-3 (day 9) surviving mice. Influenza-specific T cells were undetectable at day 5.

And the following additional text in results (page 8):-

“Using tetramers for MR1 or to immunodominant epitopes from the nucleoprotein or polymerase acidic protein of influenza virus we tracked MAIT cells and conventional antigen-specific $\alpha\beta$ T cell accumulation during PR8 infection. In MR1^{-/-} mice we observed a tendency towards decreased accumulation of antigen-specific T cells, at day 9 in splenocytes, and for nucleoprotein at day 7 in the lymph nodes (Supplementary Figure 3), suggesting MAIT cells may enhance the development of an effective antigen-specific adaptive immune response.”

References

- Liu, B., I. Mori, M. J. Hossain, L. Dong, K. Takeda and Y. Kimura (2004). "Interleukin-18 improves the early defence system against influenza virus infection by augmenting natural killer cell-mediated cytotoxicity." *J Gen Virol* 85(Pt 2): 423-428.
- Powell, T. J., T. Strutt, J. Reome, J. A. Hollenbaugh, A. D. Roberts, D. L. Woodland, S. L. Swain and R. W. Dutton (2007). "Priming with cold-adapted influenza A does not prevent infection but elicits long-lived protection against supralethal challenge with heterosubtypic virus." *J Immunol* 178(2): 1030-1038.
- Tate, M. D., H. C. Schilter, A. G. Brooks and P. C. Reading (2011). "Responses of mouse airway epithelial cells and alveolar macrophages to virulent and avirulent strains of influenza A virus." *Viral Immunol* 24(2): 77-88.
- Van Der Sluijs, K. F., L. J. Van Elden, R. Arens, M. Nijhuis, R. Schuurman, S. Florquin, J. Kwakkel, S. Akira, H. M. Jansen, R. Lutter and T. Van Der Polls (2005). "Enhanced viral clearance in interleukin-18 gene-deficient mice after pulmonary infection with influenza A virus." *Immunology* 114(1): 112-120.

REVIEWERS' COMMENTS:

Reviewer #1 (Remarks to the Author):

The revised paper by Van Wilgenburg et al has been somewhat improved by the revisions performed and additional data added. The data comparing MAIT cell adoptive transfer with NK and CD8 cell transfer is interesting, and could potentially be further investigated in more detail. However, the data now included as a supplementary figure strengthens the paper. My previous overall concern that the findings are mostly expected from previous studies remains. At the same time, this will be the first paper indicating a small but still significant protective effect of MAIT cells against a viral infection in vivo.

Given that the paper has been strengthened by the revisions and new data, I still think the authors could consider two additional points:

1) The referencing of the previously published literature is highly biased towards the authors own work. Other labs have made significant contributions to the understanding of MAIT cells in viral diseases and in lung diseases. It would be suitable to add a number of new references to acknowledge such previously published work.

2) I would suggest to make it clear already in the title that this work is based on mouse models of infection.

Reviewer #2 (Remarks to the Author):

the authors have responded appropriately to most of the concerns raised in the review of the original manuscript.

REVIEWERS' COMMENTS:

Reviewer #1 (Remarks to the Author):

The revised paper by Van Wilgenburg et al has been somewhat improved by the revisions performed and additional data added. The data comparing MAIT cell adoptive transfer with NK and CD8 cell transfer is interesting, and could potentially be further investigated in more detail. However, the data now included as a supplementary figure strengthens the paper. My previous overall concern that the findings are mostly expected from previous studies remains. At the same time, this will be the first paper indicating a small but still significant protective effect of MAIT cells against a viral infection in vivo.

Given that the paper has been strengthened by the revisions and new data, I still think the authors could consider two additional points:

1) The referencing of the previously published literature is highly biased towards the authors own work. Other labs have made significant contributions to the understanding of MAIT cells in viral diseases and in lung diseases. It would be suitable to add a number of new references to acknowledge such previously published work.

As requested by the reviewer we have inserted the following additional text and six references to MAIT cells in viral and lung diseases:-

Activation has also been observed by others in hepatitis B³, hepatitis C⁴ and Zika⁵ infections, whilst reductions in both peripheral blood MAIT cell frequencies and MAIT cell functional capacity have been observed in a number clinical studies of viral infections, including HIV⁶ and HTLV-1⁵.

Boeijen LL, Montanari NR, de Groen RA, et al. Mucosal-Associated Invariant T Cells Are More Activated in Chronic Hepatitis B, but Not Depleted in Blood: Reversal by Antiviral Therapy. *J Infect Dis* 2017;216:969-76.

Bohte FJ, O'Keefe AC, Webb LM, et al. Intra-Hepatic Depletion of Mucosal-Associated Invariant T Cells in Hepatitis C Virus-Induced Liver Inflammation. *Gastroenterology* 2017;153:1392-403 e2.

Paquin-Proulx D, Greenspun BC, Costa EA, et al. MAIT cells are reduced in frequency and functionally impaired in human T lymphotropic virus type 1 infection: Potential clinical implications. *PLoS One* 2017;12:e0175345.

Leeansyah E, Svard J, Dias J, et al. Arming of MAIT Cell Cytolytic Antimicrobial Activity Is Induced by IL-7 and Defective in HIV-1 Infection. *PLoS Pathog* 2015;11:e1005072.

We have also inserted an additional reference to MAIT cells in lung disease:

Wallington JC, Williams AP, Staples KJ, Wilkinson TMA. IL-12 and IL-7 synergize to control mucosal-associated invariant T-cell cytotoxic responses to bacterial infection. *J Allergy Clin Immunol* 2018;141:2182-95 e6.

2) I would suggest to make it clear already in the title that this work is based on mouse models of infection.

We have considered how to amend the title but overall we feel the current one is quite direct and conveys the message of the paper well. Looking at other recent papers on MAIT cells in Nature Communications (and elsewhere) that are based on mouse models, these have similarly a simple and clear title, for example those below, so overall this would be in keeping with the general style of the journal and specifically in this area.

Wang, H., C. D'Souza, et al (2018). "MAIT cells protect against pulmonary Legionella longbeachae infection." *Nat Commun* 9(1): 3350.⁷

Hegde P, Weiss E, Paradis V, et al. Mucosal-associated invariant T cells are a profibrogenic immune cell

population in the liver. Nature communications 2018;9:2146.⁸

Reviewer #2 (Remarks to the Author):

the authors have responded appropriately to most of the concerns raised in the review of the original manuscript.

We concur with the reviewer.

References

1. Dalton DK, Pitts-Meek S, Keshav S, Figari IS, Bradley A, Stewart TA. Multiple defects of immune cell function in mice with disrupted interferon-gamma genes. *Science* 1993;259:1739-42.
2. Corbett AJ, Eckle SB, Birkinshaw RW, et al. T-cell activation by transitory neo-antigens derived from distinct microbial pathways. *Nature* 2014;509:361-5.
3. Boeijen LL, Montanari NR, de Groen RA, et al. Mucosal-Associated Invariant T Cells Are More Activated in Chronic Hepatitis B, but Not Depleted in Blood: Reversal by Antiviral Therapy. *J Infect Dis* 2017;216:969-76.
4. Bolte FJ, O'Keefe AC, Webb LM, et al. Intra-Hepatic Depletion of Mucosal-Associated Invariant T Cells in Hepatitis C Virus-Induced Liver Inflammation. *Gastroenterology* 2017;153:1392-403 e2.
5. Paquin-Proulx D, Greenspun BC, Costa EA, et al. MAIT cells are reduced in frequency and functionally impaired in human T lymphotropic virus type 1 infection: Potential clinical implications. *PLoS One* 2017;12:e0175345.
6. Leeansyah E, Svard J, Dias J, et al. Arming of MAIT Cell Cytolytic Antimicrobial Activity Is Induced by IL-7 and Defective in HIV-1 Infection. *PLoS Pathog* 2015;11:e1005072.
7. Wang H, D'Souza C, Lim XY, et al. MAIT cells protect against pulmonary *Legionella longbeachae* infection. *Nature communications* 2018;9:3350.
8. Hegde P, Weiss E, Paradis V, et al. Mucosal-associated invariant T cells are a profibrogenic immune cell population in the liver. *Nature communications* 2018;9:2146.